# Prevalence and Relationships between Alexithymia, Anhedonia, Depression and Anxiety during the Belgian COVID-19 Pandemic Lockdown

**DOI:** 10.3390/ijerph192215264

**Published:** 2022-11-18

**Authors:** Farah Damerdji, Marianne Rotsaert, Benjamin Wacquier, Matthieu Hein, Gwenolé Loas

**Affiliations:** 1Department of Psychiatry & Laboratory of Psychiatric Research (ULB 266), Cliniques Universitaires de Bruxelles, Université Libre de Bruxelles (ULB), 1070 Brussels, Belgium; 2Department of Psychology & Laboratory of Psychiatric Research (ULB 266), Cliniques Universitaires de Bruxelles, Université Libre de Bruxelles (ULB), 1070 Brussels, Belgium

**Keywords:** alexithymia, anhedonia, state anhedonia, depression, anxiety, lockdown, COVID-19

## Abstract

Alexithymia and anhedonia are associated with psychiatric disorders, such as depression and anxiety. The COVID-19 pandemic lead to a significant deterioration in the mental health of the population. It is therefore important to examine the effects of lockdown on alexithymia and anhedonia and their relationships with anxiety and depression. We compared the scores and characteristics of 286 patients divided into two groups: one before lockdown (group 1, *N* = 127), the other during the progressive lockdown release (group 2, *N* = 159). The groups were homogeneous in terms of age, sex ratio, socio-professional categories, and somatic and psychiatric comorbidities. The groups were compared on the Toronto Alexithymia Scale (TAS-20) measuring alexithymia, the Beck Depression Inventory (BDI-II) measuring depression, the anhedonia subscale of the BDI-II measuring state-anhedonia and the State Trait Anxiety Inventory (STAI) measuring state and trait anxiety. The ratio of alexithymic subjects in group 1 is 22.83% to 33.33% in group 2 (*p*-value = 0.034). This suggests a significant increase in the number of alexithymic patients after lockdown. We did not observe any difference in the proportion of depressed and anxious subjects before or after lockdown. Among the different scales, higher scores were only found on the cognitive factor of alexithymia on group 2 comparatively to group 1. This study indicates an increase in the proportion of alexithymic subjects following lockdown. Unexpectedly, this was unrelated to depression, anxiety or anhedonia levels, which remained stable. Further studies are needed to confirm this result and to evaluate precisely which factors related to the lockdown context are responsible for such an increase.

## 1. Introduction

The COVID-19 pandemic and the social isolation it imposed have had an impact on the mental health of the population. Indeed, the meta-analysis published by Salari et al., shows a higher prevalence of stress (29.6%), anxiety (31.9%) and depression (33.7%) in the world population since the beginning of the pandemic [1]. The literature review by Mengin et al., also reports an increase in the prevalence of other disorders, such as addictive behaviors, sleep disorders and eating disorders following lockdown [2]. All these psychological disorders are known to have strong correlations with alexithymia [3,4].

Alexithymia is a concept developed by Sifneos and Nemiah in the early 1970s [4]. Alexithymia is defined by the difficulty to differentiate and verbalize one’s emotional states [3]. It is characterized by four main dimensions: (1) a difficulty in identifying and describing one’s feelings, (2) the presence of pragmatic, utilitarian thinking with little empathy and no imaginary content, otherwise known as “operant thinking“, (3) a decrease in the dream and fantasy content of thought and (4) a preponderant use of action to resolve conflicts [4]. Alexithymia is also a strong measure of emotion dysregulation. The prevalence of this disorder in the general population ranged from 9 to 20% [5,6] and can be assessed by different scales, the main one being the Toronto Alexithymia Scale (TAS-20) [7,8], measuring three dimensions: difficulty in identifying feelings (DIF), difficulty in describing feelings (DDF) and externally oriented thinking (EOT).

At the present time, there is no consensus regarding the stability of the alexithymic trait. While some authors consider alexithymia to be a constitutive and stable personality trait [9], others tend to refute its absolute stability and lean rather towards a relative stability of the trait [10,11], or even a variability, depending on the environment and the context of the subject [12]. The defenders of the latter position suggest that alexithymia could increase in the context of a depressive or anxious state, particularly by influencing the dimension of identification and description of feelings [12]. Lockdown could then, indirectly, through the increased incidence of depressive and anxious affects, increase alexithymia in the population. However, in a sample of 230 prisoners, Maisondieu et al., reported a high prevalence of alexithymia (42.86%) compared to the prevalence found in the general population (20%) [13]. In this study, alexithymia was associated with depression but not with the length of the prison sentence. The authors suggested that alexithymia could be a personality trait due to social isolation in prison. This suggestion was confirmed by a qualitative study which reported alexithymia traits in prisoners due to residing in prison [14].

Among the studies exploring alexithymia in COVID-19 infection, none, to our knowledge, has assessed the effect of lockdown on the prevalence of alexithymia in the population [15,16,17,18,19,20]. During the COVID-19 pandemic, several studies [15,16], using the TAS-20, reported 11% of alexithymia in the general population of the United Kingdom and 15.8% in Italian residents aged 18 years or more. Another study [17] found a high prevalence of alexithymia (76.4%) among Chinese adolescents with depression during the pandemic. Tang et al., 2020 [18], reported high levels of alexithymia associated with depression or post-traumatic stress disorder in home-quarantined Chinese university students. In their study, only DIF and DDF dimensions differentiated subjects with or without depression or post-traumatic stress disorder although EOT dimension was not significantly different. During lockdown, an online survey [19] found that higher alexithymia scores (TAS-20) were associated with increased emotional eating and another study [20] found that the DDF dimension of alexithymia was a significant predictor of the decline of quality of life during the COVID-19 lockdown relative to pre-pandemic baseline.

It is not clear on the one hand that there is an increase in the prevalence of alexithymia during the COVID-19 pandemic and on the other hand that in the event of an increase this is explained by particularities of the populations studied (age, gender, level of education).

Anhedonia—the decrease of hedonic capacity—is a symptom occurring as state or trait in several psychiatric or neurological disorders, notably depression [21]. Several studies have explored the effect of the COVID-19 pandemic on anhedonia. One study [22], in a sample of college students, reported higher anhedonia after the COVID-19 pandemic than before the pandemic. Another study [23], in cannabis users and controls, also reported increased levels of anhedonia, rated by the Snaith Hamilton Pleasure Scale (SHAPS), during lockdown. Using the SHAPS and the dimensional anhedonia rating scale, measuring state anhedonia in young healthy adults, Wellan et al., [24] did not report significant differences between pre-pandemic and during-pandemic subjects. Two studies [25,26] explored the relationships between anxiety, anhedonia and food consumption during the COVID-19 quarantine. In one study (25) anhedonia was not associated with consuming palatable foods but was associated with the low consumption of fruits or vegetables. However, the increase in serving size was positively associated with anhedonia. Another study [26] reported that subjects with severe anxiety and anhedonia, measured by the SHAPS, had increased odds of reporting an increase in bodyweight. In this study, there was no relationship between lockdown and anhedonia or the level of anxiety. In a sample of 200 post-COVID-19 patients, anhedonia, rated by the self-assessment anhedonia scale, and fatigue, rated by the fatigue assessment scale, were significantly and positively associated. Conversely, anhedonia and fatigue were significantly and negatively associated with duration after recovery [27].

As with alexithymia, the increase in anhedonia is not obvious and it is important to control the confounding variables (age, gender, etc.). Moreover, the relationship between anhedonia and alexithymia has not yet been explored in the COVID-19 pandemic.

Thus, the aim of the study was firstly to test the hypothesis that lockdown could increase the prevalence of alexithymia and/or the level of anhedonia, taking into account the confounding variables and secondly to explore the potential effect of lockdown on the relationship between alexithymia, anhedonia, anxiety or depression.

## 2. Materials and Methods

### 2.1. Participants (see Table 1)

To explore the effect of lockdown on alexithymia the clinical database of the Erasme hospital sleep laboratory was used. In Belgium, the first lockdown of the COVID-19 pandemic lasted from 18 March to 7 June in 2020. Two groups of subjects were recruited. The subjects were referred to the sleep laboratory for various complaints related to sleep, and a polysomnographic examination was performed. The study group included inpatients after the spring 2020 lockdown, i.e., between 1 June and 31 August 2020. It consists of 159 patients (66 women and 93 men) with an average age of 46.31 years. Their socioeconomic levels were: 84 (professional activity), 23 (training), 13 (retired), 10 (ill or disabled) and 29 (unemployed). The control group included inpatients hospitalized during the three months prior to the lockdown, i.e., between 1 January and 31 March 2020. It consists of 127 patients (60 women and 67 men) with an average age of 46.83 years. Their socioeconomic levels were: 80 (professional activity), 14 (training), 6 (retired), 15 (ill or disabled) and 12 (unemployed). As age, gender, educational and socioeconomic levels are known to be related to alexithymia, they have been assessed in this study [3,28,29,30]. There was no significant difference between the two groups regarding these three factors and using χ2 or Student’s *t* tests: sex ratio (χ2 = 0.94, df = 1, *p* = 0.41), age (t = 0.31, df = 284, *p* = 0.75). The socio-economic levels of the patients were assessed according to their working or non-working status and compared between the two groups. There were 120 and 39 subjects with and without activity in the study group and 100 and 27 subjects with and without activity in the control group. The difference was not significant (χ2 = 0.42, df = 1, *p* = 0.51).

This project was validated by the Erasmus-ULB ethics committee on 16 February 2021, the registration number P2020/685 was assigned on 09/12/20 (CCB reference: B4062020000307).

**Table 1 ijerph-19-15264-t001:** Demographic and psychometric characteristics of the two groups.

Variables	Study Group *N* = 159	Control Group *N* = 127	χ^2^, Fisher or *t*
**Gender (% males)**	58.5	52.8	0.34
**Age (m, sd)**	46.31 (13.7)	46.83 (14.56)	0.76
**Working status** **(% activity)**	75.5	78.7	0.77
**Psychiatric comorbidity**	20.7	15.7	0.29
**Somatic comorbidity**	99.4	96.8	0.17
**Alexithymia (%)**	33.3	22.8	**0.034**
**Depression (%)**	48.4	51.2	0.37
**State-anxiety (%)**	59.7	59.8	0.54
**Trait-anxiety (%)**	57.9	52.8	0.23
**DIF**	15.7 (6.83)	15.03 (5.96)	0.19
**DDF**	12.85 (5.05)	12.61 (4.53)	0.34
**EOT**	18.7 (4.56)	17.8 (4.48)	**0.046**
**TAS-20**	47.25 (13.07)	45.44 (11.61)	0.11
**BDI-II**	13.07 (9.72)	13.32 (10.15)	0.42
**ANH-BDI**	1.85 (1.86)	1.94 (192)	0.33
**STAI-State**	50.38 (12.03)	50.53 (11.65)	0.46
**STAI-Trait**	48.14 (11.57)	48.74 (12.05)	0.34

Toronto Alexithymia Scale (TAS-20); difficulty in identifying feelings (DIF); difficulty in describing feelings (DDF); externally oriented thinking (EOT); Beck Depression Inventory-II (BDI-II); anhedonia subscale (ANH-BDI) of the Beck Depression Inventory-II; State Trait Anxiety Inventory (STAI) measuring state (STAI-State) and trait anxiety (STAI-Trait). In bold face: *p* ≤ 0.05.

### 2.2. Measures (see Table 1)

Alexithymia was assessed by the French version of the TAS-20 [7,8]. This scale is currently the most commonly used. It is a self-report scale with 20 items scored between 1 and 5 for a total of 20 to 100. It includes three subscales: difficulty in identifying feelings (DIF), difficulty in describing feelings (DDF) and externally oriented thinking (EOT). A total score greater than or equal to 56 indicates the presence of alexithymia [31].

State anhedonia was rated by the anhedonia subscale (ANH-BDI) of the Beck Depression Inventory-II (BDI-II) [32,33] that contains three items (Item 4 or Loss of Pleasure (LP) ‘I can’t get any pleasure from the things I used to enjoy’; item 12 or Loss of Interest (LI) ‘It’s hard to get interested in anything’ and item 21 or Loss of Interest in Sex (LIS) ‘I have lost interest in sex completely’).

Depression was assessed using the French version of the BDI-II [32,33]. It is a 21-item scale, each item scored between 0 and 3. A BDI score greater than or equal to 12 indicates the presence of depression [33].

Anxiety was assessed using the French version of State-Trait Anxiety Inventory (STAI) [34,35]. STAI includes 20 items assessing state anxiety and 20 others assessing trait anxiety, each scored between 1 and 4. The score of each subscale varies between 20 and 80 and indicates the level of anxiety. Scores higher than 45 [34] indicate high level of state or trait anxiety.

The French versions of these scales have been validated in both healthy and psychiatric subjects [8,33,35].

### 2.3. Statistical Analyses

For the categorical analysis, we considered a standard threshold above which the subject is categorized alexithymic (TAS-20 total ≥ 56) [31], depressive (BDI-II ≥ 12) [33] or anxious (STAI-State or STAI-Trait ≥ 45) [35]. The prevalence of alexithymia, depression and anxiety was expected higher in the study group than in the control group and thus was compared using unilateral Fisher exact test.

To exclude them as confounding factors, psychiatric and somatic comorbidities were compared between the two groups using Fisher exact tests.

For dimensional analysis, the mean scores of the various scales and subscales were compared between the two groups using unilateral Student’s *t* tests. It was expected that significantly higher scores would be found on the TAS-20, ANH-BDI, STAI and BDI-II for the study group compared with the control group. In each group, the Pearson’s correlation between the rating scales was calculated.

## 3. Results

### 3.1. Categorical Analyses (see Table 1)

The prevalence of alexithymia was higher in the study group (33.33%) than in the control group (22.83%). The difference was statistically significant (*p* = 0.034). The prevalence of depression did not significantly differ between the two groups (study group: 48.43% control group: 51.18%) (*p* = 0.37). The prevalence of state anxiety did not significantly differ between the two groups (study group: 59.75%; control group: 59.84%) (*p* = 0.54). The prevalence of the anxiety trait did not significantly differ between the two groups (study group: 57.86%; control group: 52.76%) (*p* = 0.23). There was no statistically significant difference in comorbidities between the two groups, neither psychiatric (*p* = 0.29) nor somatic (*p* = 0.17).

### 3.2. Dimensional Analyses (see Table 1 and Table 2)

Comparison between the scores of the alexithymia subscales DIF (control group: m = 15.03, sd = 5.96; study group: m = 15.7, sd = 6.83) and DDF (control group: m = 12.61, sd = 4.53; study group: m = 12.85, sd = 5.05) showed no statistically significant difference between the two groups (DIF: t = −0.87, df = 284, *p* = 0.193; DDF: t = −0.41, df = 284, *p* = 0.341). The EOT subscale showed a significant difference (study group: m = 18.70, sd = 4.56; control group: m = 17.79, sd = 4.48) (t = −1.69, df = 284, *p* = 0.046). There is no significant difference between the two groups for the scores of BDI-II, ANH-BDI, STAI-State or STAI-Trait.

In each group significant correlations were observed for DDF and DIF subscales and the anxiety, depression and anhedonia scales although EOT subscale correlated significantly only with the DDF, DIF or total TAS-20 scales. In the study group, the values of the correlations between EOT and DIF or DDF were lower than the values found in the control group (see Table 2).

**Table 2 ijerph-19-15264-t002:** Correlations between the rating scales (bottom: study group; below: control group).

	DIF	DDF	EOT	TAS-20	ANH-BDI	BDI-II	STAI-State	STAI-Trait
**DIF**	1.00	**0.69**	**0.22**	**0.87**	**0.39**	**0.63**	**0.53**	**0.56**
**DDF**	**0.51**	1.00	**0.36**	**0.87**	**0.18**	**0.40**	**0.46**	**0.46**
**EOT**	**0.28**	**0.41**	1.00	**0.6**	0.07	0.03	0.01	0.03
**TAS-20**	**0.82**	**0.81**	**0.69**	1.00	**0.3**	**0.49**	**0.46**	**0.48**
**ANH-BDI**	**0.42**	**0.25**	0.16	**0.38**	1.00	**0.72**	**0.33**	**0.42**
**BDI-II**	**0.47**	**0.38**	0.13	**0.44**	**0.72**	1.00	**0.63**	**0.72**
**STAI-State**	**0.38**	**0.2**	0.09	**0.31**	**0.33**	**0.67**	1.00	**0.66**
**STAI-Trait**	**0.37**	**0.21**	0.08	**0.48**	**0.3**	**0.6**	**0.54**	1.00

Toronto Alexithymia Scale (TAS-20); difficulty in identifying feelings (DIF); difficulty in describing feelings (DDF); externally oriented thinking (EOT); Beck Depression Inventory-II (BDI-II); anhedonia subscale (ANH-BDI) of the Beck Depression Inventory-II; State Trait Anxiety Inventory (STAI) measuring state (STAI-State) and trait anxiety (STAI-Trait). In bold face: *p* ≤ 0.05.

## 4. Discussion

The main result of the present study is the increase of alexithymia among the subjects evaluated after the lockdown and compared with the subjects evaluated before the lockdown. This increase is observed in categorical as well as in dimensional analyses. The prevalence of alexithymia is 33.33% after lockdown and 22.83% before lockdown. In dimensional analysis only the score of the EOT subscale increased after lockdown. This increase was not explained by the potential effect of socio-demographical variables as the two groups did not differ for gender, age or socio-economic status. However, the increase was not explained by psychiatric or somatic comorbidities either as the two groups did not differ on the prevalence of these comorbidities.

The increase in alexithymia of nearly 10% is in contradiction with the hypothesis that alexithymia is a stable personality trait. Indeed, for some authors, alexithymia is a stable construct over time [9,10,11]. Honkalampi et al., support variability [12], as alexithymia fluctuates according to the depressive state, until it disappears after the treatment and resolution of the depression. The literature widely reports strong links between alexithymia, depression and anxiety [3,12]. In this study, the increase of the prevalence of alexithymia was not explained by depression or anxiety, as the prevalence of depression or anxiety did not differ between the two groups. However, in dimensional analysis, except for the EOT factor, there was no significant difference between the groups for the depression, anxiety or alexithymia subscales. There was no significant difference for anhedonia-state between the two groups.

A potential explanation for this result could be the fact that anxiety, depression and anhedonia levels were investigated not during lockdown but during progressive lockdown release. Indeed, the study by Cecchetto et al., shows a clear attenuation of negative emotions during the lockdown lifting phase and the easing of the sanitary measures [19]. Another explanation for the stability of anxious and depressive affect between the two groups could be that the decrease in introspection, assessed by the EOT scale, regulates anxiety and depression in the confined population. Indeed, regarding the stability of anxiety, Osimo et al., established a link between the increase in the EOT subscale and the decrease of anxiety during lockdown [16]. The authors explain this phenomenon by the fact that a decrease in introspection could, by keeping negative affects at bay, be a temporary manner to adapt to a difficult situation.

Externally oriented thinking, contrary to the other dimensions of alexithymia, was not related to anxiety, depression or anhedonia in various groups of subjects [36,37]. In a sample of undergraduate students, Motan and Gencoz [38] reported that only the dimensions of DIF and DDF were associated with anxiety or depression. Individual differences related to emotion were described by Gohm and Clore [39] in five conceptual categories: absorption, attention, clarity, intensity and expression. The EOT subscale has been included in attention category and DDF and DIF have been included in the clarity category, defined as the ability to know what one is feeling and to distinguish one’s emotions from one another.

Our results are in line with this explanation. Indeed, it is notable that of the three alexithymia subscales, only the EOT shows an increase, with the DIF and DDF remaining stable. Taking into account that EOT is the cognitive component of alexithymia and that DIF and DDF are the emotional components of alexithymia, our results could support the hypothesis that alexithymia is a defense mechanism against major anxiety related to the global context of the pandemic, which would partly explain its increase [40].

Another explanation is that because during lockdown, increased levels of loneliness were observed, the level of alexithymia was consequently also increased. Several studies have shown strong relationships between loneliness and alexithymia [41]. In university students, significant correlations were reported between DIF, DDF and the three components of loneliness (romantic, family and social) although EOT correlated significantly with romantic loneliness [41]. During the COVID-19 lockdown, it was reported that emotional dysregulation levels partially mediated the longitudinal relationship between loneliness and both depression and stress [42]. It has been extensively investigated and reported that the processing of emotional stimuli and inhibitory functions in alexithymic individuals are altered [43,44] and it can be suggested that the social consequences of the COVID-19 lockdown could have increased the process dysfunction.

The last explanation is that lockdown and social distancing have increased affective flattening and externally oriented thinking. Thus, alexithymia could be a consequence of lockdown. Distancing oneself from one’s emotions and those of others would thus be a way for the subject to deal with an overwhelming situation.

In addition to being a threat to individual health, alexithymia may also have sociological and societal implications [45]. On the one hand, alexithymic subjects are more prone to social isolation due to the relational consequences of this disorder [3]. On the other hand, the literature shows a strong link between emotions and moral reasoning [3]. Thus, distancing oneself from one’s emotions and those of others, as well as displaying a weak introspective capacity, would impact moral decision-making. This is reminiscent of Hannah Arendt’s concept of the “banality of evil”, coined after the trial of the SS officer Adolf Eichmann, considered by Sifneos to be an alexithymic personality [45].

On another level, we observe following the lockdown, a democratization of home remote working in the professional sphere. One could then imagine that a geographical and emotional distance between the employer and the employee imposed by this new practice together with an increase in alexithymia could represent an additional risk for suffering at work. It would be interesting to conduct studies on the impact of home working on alexithymia among managers, employees and their hierarchical relationships.

### Strengths and Limitations of the Study

Alexithymia was assessed during progressive lockdown release and not during the lockdown period, which may have influenced the results, particularly regarding anxious–depressive affects. This study controlled for the economic level but not for the level of education directly.

Considering the growing consensus towards a stable alexithymic trait, the increase in the ratio of alexithymic patients could be due to unknown confounding factors that we did not verify.

However, this work was carried out in the unusual context of a lockdown, offers a new opportunity to study the variability of alexithymia. Moreover, this study, by supporting the variability of alexithymia, invites research on potential therapeutic means [10].

## 5. Conclusions

Our study allows us to establish a significant increase in the proportion of alexithymic patients in the population following lockdown. This tendency is even more striking, on the one hand, because it concerns a trait considered as inherent to one’s personality and therefore not particularly variable in time and, on the other hand, because it seems to be independent of anxiety and depression. This tendency for alexithymia to increase following lockdown raises again the issue of the stability of alexithymia. This suggests that alexithymia may be influenced by factors related to lockdown, such as social isolation or global anxiety when facing an unfamiliar infectious disease or the wearing of a mask which does not facilitate the reading of emotions. Further studies are needed to support the variability in the proportion of alexithymic subjects following lockdown and uncover what the exact factors are. It would be interesting to study either the influence of the environment (city versus countryside) [46] or to test the hypothesis that specific groups of subjects are, more vulnerable such as healthcare professionals [47] or older people with physical disabilities [48]. Accepting the variability of the trait also implies the possibility of taking action and of treating alexithymic patients [10]. The exact factors on which to act and how this can be carried out should be defined by future research.

## Data Availability

Requests for access will be reviewed by the corresponding author.

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
