# Peer review of "Prevalence and Relationships between Alexithymia, Anhedonia, Depression and Anxiety during the Belgian COVID-19 Pandemic Lockdown"

_ijerph, 2022, doi:10.3390/ijerph192215264_

Round 1
Reviewer 1 Report
1) The manuscript does not show some key information of the participants, such as age, gender, education level, income, etc.
2) Furthermore, the model construction and conclusion in this manuscript ignore the socio-economic information of participants, which should be discussed.
3) In the comparison between the experimental group and the control group, only alexithymia has a significant difference, while other factors have no significant difference. Why? More convincing statements are needed in the discussion section.
4) Some key references are omitted in the article, and it is suggested to add them. For example:
- Jato Espino Daniel,Moscardó Vanessa,Rodríguez Alejandro Vallina & Lázaro Esther.(2021).Spatial statistical analysis of the relationship between self-reported mental health during the COVID-19 lockdown and closeness to green infrastructure. Urban Forestry & Urban Greening(prepublish). doi:10.1016/J.UFUG.2021.127457.
- Ayalew Mohammed,Deribe Bedilu,Abraham Yacob,Reta Yared,Tadesse Fikru,Defar Semira... & Ashegu Tebeje.(2021).Prevalence and determinant factors of mental health problems among healthcare professionals during COVID-19 pandemic in southern Ethiopia: multicentre cross-sectional study.. BMJ open(12). doi:10.1136/BMJOPEN-2021-057708.
- Jia Ziyu,Xu Shijia,Zhang Zican,Cheng Zhengyu,Han Haoqing,Xu Haoxiang... & Zhou Zhengxu.(2021).Association between mental health and community support in lockdown communities during the COVID-19 pandemic: Evidence from rural China. Journal of Rural Studies(prepublish). doi:10.1016/J.JRURSTUD.2021.01.015.
- Steptoe Andrew & Di Gessa Giorgio.(2021).Mental health and social interactions of older people with physical disabilities in England during the COVID-19 pandemic: a longitudinal cohort study. The Lancet Public Health(6). doi:10.1016/S2468-2667(21)00069-4.
Author Response
Responses to the reviewer 1 1) Age, gender, professional activity are given in the text and in the Table 1 (see lines 128-141) 2) Socio-economic information of the participants is a potential confounding variable that has been controled (see lines 138-142). 3) The discussion has been increased (lines 276-286) and the effect of loneliness on alexithymia is now discussed. 4) 3 references suggested by the reviewers has been added (46-48) and cited lines330-333.
Reviewer 2 Report
Dear Editor,
in the manuscript entitled “Prevalence and Relationships Between Alexithymia, Anhedonia, Depression and Anxiety During the Belgian Lockdown Covid-19 Pandemic”, the authors compared two groups’ scores measured from TAS-20, ANH-BDI, STAI and BDI-II (Group1 - before lockdown - N = 127; Group 2 during the progressive lockdown release - N = 159). Results showed a significant increase in the number of alexithymic patients after lockdown (according to the measured TAS20 scores).
This is an interesting study. However, for what concerns the manuscript I have found some points not addressed. In my view the manuscript is not ready for publication yet.
These are my comments:
1. The introduction is very redundant and not very smooth. I believe that authors should make an effort to make the text more readable, clear and enjoyable by a wide audience such as that of IJERPH.
2. Overall I believe that there is a big limitation in this study. Education and the socioeconomic status are not controlled between the two groups, but it has been proven that it there is a relationship between the alexithymic trait with the years of education and with the socioeconomic status (e.g. Lane et al., 1998).
Authors should add in the limitation section of the discussion this limitation.
Lane RD, Sechrest L, Riedel R. Sociodemographic correlates of alexithymia. Compr Psychiatry. 1998 Nov-Dec;39(6):377-85
3. Emotion dysregulation in alexithymia has been extensively investigated, with compelling evidence that the processing of emotional stimuli and inhibitory functions in alexithymic individuals are altered (e.g. Van der velde et al., 2013; Gavazzi et al., 2017). Notably during the covid-19 lockdown it has been reported how much dysregulation may affect other clinical traits (Velotti et al., 2019). I believe that the authors may improve their paper discussing the relationship between the measured alteration of brain activation of inhibitory control circuit in alexithymic patients (e.g. Van der Velde et al., 2013; Gavazzi et al., 2017) and the increment of dysregulation observed during the covid lockdown by taking into consideration the significant increase in the number of alexithymic patients they have measured after the lockdown.
I believe that this manuscript may benefit from a discussion of these relationships; in particular, one may suggest future studies to investigate this relationship or propose some speculations.
Gavazzi G, Orsolini S, Rossi A, Bianchi A, Bartolini E, Nicolai E, Soricelli A, Aiello M, Diciotti S, Viggiano MP, Mascalchi M. Alexithymic trait is associated with right IFG and pre-SMA activation in non-emotional response inhibition in healthy subjects. Neurosci Lett. 2017 Sep 29;658:150-154.
Van der Velde, M.N. Servaas, K.S. Goerlich, R. Bruggeman, P. Horton, S.G. Costafreda, A. Aleman, Neural correlates of alexithymia: a meta-analysis of emotion processing studies, Neurosci. Biobehav. Rev. 37 (2013) 1774–1785.
Velotti P, Rogier G, Beomonte Zobel S, Castellano R, Tambelli R. Loneliness, Emotion Dysregulation, and Internalizing Symptoms During Coronavirus Disease 2019: A Structural Equation Modeling Approach. Front Psychiatry. 2021 Jan 8;11:581494. doi: 10.3389/fpsyt.2020.581494. PMID: 33488417; PMCID: PMC7819957.
Author Response
Responses to the reviewer 2 1) In the introduction several sentences summarizes the literature exploring alexithymia or anhedonia in COVID-19 pandemic. The hypothese are more clearly presented. 2) The effect of socio-economic status is controled (see lines 141). See reference 29 and we have added a new reference (ref 30, Lane et al). As professional activity is not a direct measure of educational status this limitation has been mentioned (see lines 311-312). 3) We have followed the suggestion of the reviewer in the discussion presenting a new explanation of our results (effect of loneliness..) see lines 276-286. Four references were added (41-44).Round 2
Reviewer 2 Report
I believe that the paper is ready to be published.